# INTERVENTION GENERATIVE ADVERSARIAL NETS

## ABSTRACT

In this paper we propose a novel approach for stabilizing the training process of Generative Adversarial Networks as well as alleviating the mode collapse problem. The main idea is to incorporate a regularization term that we call *intervention* into the objective. We refer to the resulting generative model as *Intervention Generative Adversarial Networks* (IVGAN). By perturbing the latent representations of real images obtained from an auxiliary encoder network with Gaussian invariant interventions and penalizing the dissimilarity of the distributions of the resulting generated images, the intervention term provides more informative gradient for the generator, significantly improving training stability and encouraging mode-covering behaviour. We demonstrate the performance of our approach via solid theoretical analysis and thorough evaluation on standard real-world datasets as well as the stacked MNIST dataset.

## 1 INTRODUCTION

As one of the most important advances in generative models in recent years, Generative Adversarial Networks (GANs) (Goodfellow et al., 2014) have been attracting great attention in the machine learning community. GANs aim to train a generator network that transforms simple vectors of noise to produce "realistic" samples from the data distribution. In the basic training process of GANs, a discriminator and a target generator are trained in an adversarial manner. The discriminator tries to distinguish the generated fake samples from the real ones, and the generator tries to fool the discriminator into believing the generated samples to be real.

Though successful, there are two major challenges in training GANs: the instability of the training process and the mode collapse problem. To deal with these problems, one class of approaches focus on designing more informative objective functions (Salimans et al., 2016; Mao et al., 2016; Kodali et al., 2018; Arjovsky & Bottou; Arjovsky et al., 2017; Gulrajani et al., 2017; Zhou et al., 2019). For example, Mao et al. (2016) proposed *Least Squares GAN* (LSGAN) that uses the least squares loss to penalize the outlier point more harshly. Arjovsky & Bottou discussed the role of the Jensen-Shannon divergence in training GANs and proposed WGAN (Arjovsky et al., 2017) and WGAN-GP (Gulrajani et al., 2017) that use the more informative Wasserstein distance instead. Other approaches enforce proper constraints on latent space representations to better capture the data distribution (Makhzani et al., 2015; Larsen et al., 2015; Che et al., 2016; Tran et al., 2018). A representative work is the *Adversarial Autoencoders* (AAE) (Makhzani et al., 2015) which uses the discriminator to distinguish the latent representations generated by encoder from Gaussian noise. Larsen et al. (2015) employed image representation in the discriminator as the reconstruction basis of a VAE. Their method turns pixel-wise loss to feature-wise, which can capture the real distribution more simply when some form of invariance is induced. Different from VAE-GAN, Che et al. (2016) regarded the encoder as an auxiliary network, which can promote GANs to pay much attention on missing mode and derive an objective function similar to VAE-GAN. A more detailed discussion of related works can be found in Appendix C.

In this paper we propose a novel technique for GANs that improve both the training stability and the quality of generated images. The core of our approach is a regularization term based on the latent representations of real images provided by an encoder network. More specifically, we apply auxiliary intervention operations that preserve the standard Gaussian (e.g., the noise distribution) to these latent representations. The perturbed latent representations are then fed into the generator to produce *intervened* samples. We then introduce a classifier network to identify the right intervention operations that would have led to these intervened samples. The resulting negative cross-entropy loss

is added as a regularizer to the objective when training the generator. We call this regularization term the *intervention loss* and our approach *InterVention Generative Adversarial Nets* (IVGAN).

We show that the intervention loss is equivalent with the JS-divergence among multiple intervened distributions. Furthermore, these intervened distributions interpolate between the original generative distribution of GAN and the data distribution, providing useful information for the generator that is previously unavailable in common GAN models (see a thorough analysis on a toy example in Example 1). We show empirically that our model can be trained efficiently by utilizing the parameter sharing strategy between the discriminator and the classifier. The models trained on the MNIST, CIFAR-10, LSUN and STL-10 datasets successfully generate diverse, visually appealing objects, outperforming state-of-the-art baseline methods such as WGAN-GP and MRGAN in terms of the *Frèchet Inception Distance* (FID) (proposed in (Heusel et al., 2017)). We also perform a series of experiments on the stacked MNIST dataset and the results show that our proposed method can also effectively alleviate the mode collapse problem. Moreover, an ablation study is conducted, which validates the effectiveness of the proposed intervention loss.

In summary, our work offers three major contributions as follows. (i) We propose a novel method that can improve GAN's training as well as generating performance. (ii) We theoretically analyze our proposed model and give insights on how it makes the gradient of generator more informative and thus stabilizes GAN's training. (iii) We evaluate the performance of our method on both standard real-world datasets and the stacked MNIST dataset by carefully designed expriments, showing that our approach is able to stabilize GAN's training and improve the quality and diversity of generated samples as well.

## 2 PRELIMINARIES

**Generative adversarial nets**  The basic idea of GANs is to utilize a discriminator to continuously push a generator to map Gaussian noise to samples drawn according to an implicit data distribution. The objective function of the vanilla GAN takes the following form:

$$\min_G \max_D \left\{ V(D, G) \triangleq \mathbb{E}_{x \sim p_{data}} \log(D(x)) + \mathbb{E}_{z \sim p_z} \log(1 - D(G(z))) \right\}, \tag{1}$$

where $p_z$ is a prior distribution (e.g., the standard Gaussian). It can be easily seen that when the discriminator reaches its optimum, that is, $D^*(x) = \frac{p_{data}(x)}{p_{data}(x) + p_G(x)}$, the objective is equivalent to the Jensen-Shannon (JS) divergence between the generated distribution $p_G$ and data distribution $p_{data}$:

$$JS(p_G \| p_{data}) \triangleq \frac{1}{2} \left\{ KL(p_G \| \frac{p_G + p_{data}}{2}) + KL(p_{data} \| \frac{p_G + p_{data}}{2}) \right\}.$$

Minimizing this JS divergence guarantees that the generated distribution converges to the data distribution given adequate model capacity.

**Multi-distribution JS divergence**  The JS divergence between two distributions $p_1$ and $p_2$ can be rewritten as

$$JS(p_1 \| p_2) = H(\frac{p_1 + p_2}{2}) - \frac{1}{2}H(p_1) - \frac{1}{2}H(p_2),$$

where $H(p)$ denotes the entropy of distribution $p$. We observe that the JS-divergence can be interpreted as the entropy of the mean of the two distribution minus the mean of two distributions' entropy. So it is immediate to generalize the JS-divergence to the setting of multiple distributions. In particular, we define the JS-divergence of $p_1, p_2, \ldots, p_n$ with respect to weights $\pi_1, \pi_2, \ldots, \pi_n$ ($\sum \pi_i = 1$ and $\pi_i \geq 0$) as

$$JS_{\pi_1, \ldots, \pi_n}(p_1, p_2, \ldots, p_n) \triangleq H(\sum_{i=1}^n \pi_i p_i) - \sum_{i=1}^n \pi_i H(p_i). \tag{2}$$

The two-distribution case described above is actually a special case of the 'multi-JS divergence', where $\pi_1 = \pi_2 = \frac{1}{2}$. When $\pi_i > 0 \ \forall i$, it can be found immediately by Jensen's inequality that $JS_{\pi_1, \ldots, \pi_n}(p_1, p_2, \ldots, p_n) = 0$ if and only if $p_1 = p_2 = \cdots = p_n$.

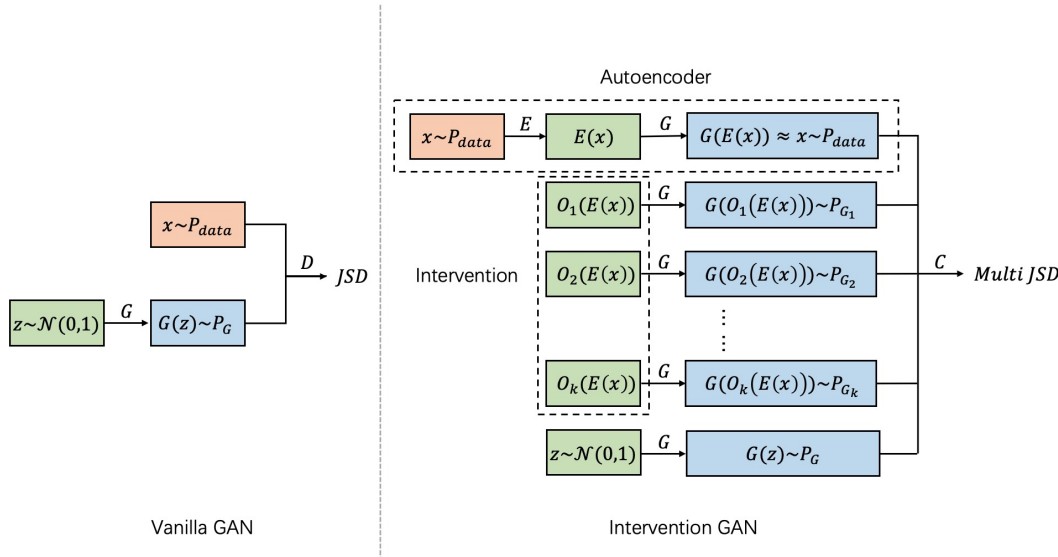

Figure 1: Comparison between the vanilla GAN loss and the Intervention loss. Here the intervened samples are generated through different intervention operations, namely $O_1, ..., O_k$.

## 3 METHODOLOGY

Training GAN has been challenging, especially when the generated distribution and the data distribution are far away from each other. In such cases, the discriminator often struggles to provide useful information for the generator, leading to instability and mode collapse problems. The key idea behind our approach is that we construct auxiliary intermediate distributions that interpolate between the generated distribution and the data distribution. To do that, we first introduce an encoder network and combine it with the generator to learn the latent representation of real images within the framework of a standard autoencoder. We then perturb these latent representations with carefully designed intervention operations before feeding them into the generator to create these auxiliary interpolating distributions. A classifier is used to distinguish the intervened samples, which leads to an intervention loss that penalizes the dissimilarity of these intervened distributions. The reconstruction loss and the intervention loss are added as regularization terms to the standard GAN loss for training. We start with an introduction of some notation and definitions.

**Definition 1** (Intervention). *Let $O$ be a transformation on the space of $d$-dimension random vectors and $\mathbb{P}$ be a probability distribution whose support is in $\mathbb{R}^d$. We call $O$ a $\mathbb{P}$-intervention if for any $d$-dimensional random vector $X$, $X \sim \mathbb{P} \Rightarrow O(X) \sim \mathbb{P}$.*

Since the noise distribution in GAN models is usually taken to be standard Gaussian, we use the standard Gaussian distribution as the default choice of $\mathbb{P}$ and abbreviate the $\mathbb{P}$-intervention as *intervention*, unless otherwise claimed. One of the simplest groups of interventions is **block substitution**. Let $Z \in \mathbb{R}^d$ be a random vector, $k \in \mathbb{N}$ and $k|d$. We slice $Z$ into $k$ blocks so that every block is in $\mathbb{R}^{\frac{d}{k}}$. A block substitution intervention $O_i$ is to replace the $i$th block of $Z$ with Gaussian noise, $i = 1, \ldots, \frac{d}{k}$. We will use block substitution interventions in the rest of the paper unless otherwise specified. Note that our theoretical analysis as well as the algorithmic framework do not depend on the specific choice of the intervention group.

**Notation** We use $E, G, D, f$ to represent encoder, generator, discriminator and classifier, respectively. Here and later, $p_{real}$ means the distribution of the real data $X$, and $p_z$ is the prior distribution of noise $z$ defined on the latent space (usually is taken to be Gaussian). Let $O_i, i = 1, \ldots, k$ denote $k$ different interventions and $p_i$ be the distribution of intervened sample $X_i$ created from $O_i$ (namely $X_i = G(O_i(E(X)))$).

**Intervention loss** The intervention loss is the core of our approach. More specifically, given a latent representation $z$ that is generated by an encoder network $E$, we sample an intervention $O_i$ from a complete group $S = \{O_1, \ldots, O_k\}$ and obtain the corresponding intervened latent variable $O_i(z)$ with label $e_i$. These perturbed latent representations are then fed into the generator to produce

*intervened* samples. We then introduce an auxiliary classifier network to identify which intervention operations may lead to these intervened samples. The intervention loss $\mathcal{L}_{IV}(G, E)$ is simply the resulting negative cross-entropy loss and we add that as a regularizer to the objective function when training the generator. As we can see, the intervention loss is used to penalize the dissimilarity of the distributions of the images generated by different intervention operations. Moreover, it can be noticed that the classifier and the combination of the generator and the encoder are playing a two-player adversarial game and we will train them in an adversarial manner. In particular, we define

$$\mathcal{L}_{IV}(G, E) = -\min_f V_{class}, \quad \text{where} \quad V_{class} = \mathbb{E}_{i \sim \mathcal{U}([k])} \mathbb{E}_{x' \sim p_i} - e_i^{\mathrm{T}} \log f(x'). \quad (3)$$

**Theorem 1** (Optimal Classifier). *The optimal solution of the classifier is the conditional probability of label $y$ given $X'$, where $X'$ is the intervened sample generated by the intervention operation sampled from $S$. And the minimum of the cross entropy loss is equivalent with the negative of the Jensen Shannon divergence among $\{p_1, p_2, ..., p_k\}$. That is,*

$$f_i^*(x) = \frac{p_i(x)}{\sum_{j=1}^k p_j(x)} \quad and \quad \mathcal{L}_{IV}(G, E) = JS(p_1, p_2, ..., p_k) + Const. \quad (4)$$

The proof can be found in Appendix A.1. Clearly, the intervention loss is an approximation of the multi-JS divergence among the intervened distributions $\{p_i : i \in [k]\}$. If the intervention reaches its global minimum, we have $p_1 = p_2 = \cdots = p_k$. And it reaches the maximum $\log k$ if and only if the supports of these $k$ distributions do not intersect with each other. This way, the probability that the 'multi' JS-divergence has constant value is much smaller, which means the phenomenon of gradient vanishing should be rare in IVGAN. Moreover, as shown in the following example, due to these auxiliary intervened distributions, the intervention is able to provide more informative gradient for the generator that is not previously available in other GAN variants.

**Example 1** (Square fitting). *Let $X_0$ be a random vector with distribution $\mathcal{U}(\alpha)$, where $\alpha = [-\frac{1}{2}, \frac{1}{2}] \times [-\frac{1}{2}, \frac{1}{2}]$. And $X_1 \sim \mathcal{U}(\beta)$, where $\beta = [a - \frac{1}{2}, a + \frac{1}{2}] \times [\frac{1}{2}, \frac{3}{2}]$ and $0 \le a \le 1$. Assuming we have a perfect discriminator (or classifier), we compute the vanilla GAN loss (i.e. the JS-divergence) and the intervention loss between these two distributions, respectively,*

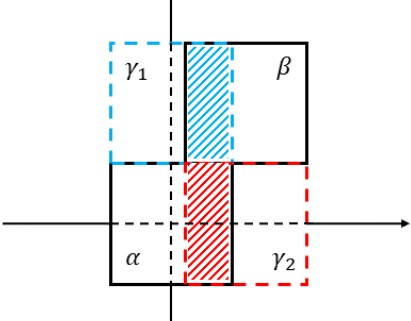

- *$JS(X_0 \| X_1) = \log 2$.*

- *In order to compute the intervention loss we need figure out two intervened samples' distributions evolved from $\mathcal{U}(\alpha)$ and $\mathcal{U}(\beta)$. $Y_1 \sim \mathcal{U}(\gamma_1)$; $\gamma_1 = [-\frac{1}{2}, \frac{1}{2}] \times [\frac{1}{2}, \frac{3}{2}]$ and $Y_2 \sim \mathcal{U}(\gamma_2)$; $\gamma_2 = [a - \frac{1}{2}, a + \frac{1}{2}] \times [-\frac{1}{2}, \frac{1}{2}]$. Then the intervention loss is the multi JS-divergence among these four distributions:*

Figure 2: The supports of the two original distribution are the squares with black border, and the supports of the synthetic distributions are the area enclosed by red and blue dotted line, respectively.

$$\mathcal{L}_{IV} = JS(X_0; X_1; Y_1; Y_2)$$
$$= -\int_{A^c} \frac{1}{4} \log \frac{1}{4} d\mu - \int_A \frac{1}{2} \log \frac{1}{2} d\mu - H(X_0) = \frac{\log 2}{2} [\mu(A^c) + \mu(A)]$$
$$= \frac{\log 2}{2} \times 2(2 - a) - H(X_0) = -(\log 2)a - Const.$$

*Here $A$ is the shaded part in Figure 2 and $A^c = \{\alpha \cup \beta \cup \gamma_1 \cup \gamma_2\} \backslash A$. The most important observation is that the intervention loss is a function of parameter $a$ and the traditional GAN loss is always constant. When we replace the JS with other $f$-divergence, the metric between $\mathcal{U}(\alpha)$ and $\mathcal{U}(\beta)$ would still remain constant. Hence in this situation, we can not get any information from standard JS for training of the generator but the intervention loss works well.*

---

**Algorithm 1** Intervention GAN

---

**Input** learning rate $\alpha$, regularization parameters $\lambda$ and $\mu$, dimension $d$ of latent space, number $k$ of blocks in which the hidden space is divided, minibatch size $n$, Hadamard multiplier $*$

1: **for** number of training iterations **do**
2:      Sample minibatch $z_j$, $j = 1, ..., n$, $z_j \sim p_z$
3:      Sample minibatch $x_j$, $j = 1, ..., n$, $x_j \sim p_{real}$
4:      **for** number of inner iteration **do**
5:          $w_j \leftarrow E(x_j)$, $j = 1, ..., n$
6:          Sample Gaussian noise $\epsilon$
7:          Sample $i_j \in [k]$, $j = 1, ..., n$
8:          $x'_j \leftarrow G(O_{i_j}(w_j))$
9:          Update the parameters of $D$ by:
10:          $\theta_D \leftarrow \theta_D - \frac{\alpha}{2n}\nabla_{\theta_D}\mathcal{L}_{adv}(\theta_D)$
11:          Update the parameters of $f$ by:
12:          $\theta_f \leftarrow \theta_f + \frac{\alpha}{n}\nabla_{\theta_f} \sum_{j=1}^{n} \log f_{i_j}(x'_j)$
13:          Calculate $\mathcal{L}_{Adv}$ and $\hat{\mathcal{L}}_{IV}$
14:      Update the parameter of $G$ by:
15:          $\theta_G \leftarrow \theta_G + \frac{\alpha}{n}\nabla_{\theta_G} \left\{ \hat{\mathcal{L}}_{Adv} + \lambda\hat{\mathcal{L}}_{recon} + \mu\hat{\mathcal{L}}_{IV} \right\}$
16:      Update the parameter of $E$ by:
17:          $\theta_E \leftarrow \theta_E + \frac{\alpha}{n}\nabla_{\theta_E} \left\{ \lambda\hat{\mathcal{L}}_{recon} + \mu\hat{\mathcal{L}}_{IV} \right\}$

---

**Reconstruction loss**    In some sense we expect our encoder to be a reverse function of the generator. So it is necessary for the objective function to have a term to push the map composed of the Encoder and the Generator to have the ability to reconstruct the real samples. Not only that, we also hope that the representation can be reconstructed from samples in the pixel space.

Formally, the reconstruction loss can be defined by the $\ell_p$-norm ($p \geq 1$) between the two samples, or in the from of the Wasserstein distance between samples if images are regarded as a histogram. Here we choose to use the $\ell_1$-norm as the reconstruction loss:

$$\mathcal{L}_{recon} = \mathbb{E}_{X \sim p_{real}} \|G(E(X)) - X\|_1 + \mathbb{E}_{i \sim \mathcal{U}([k])} \mathbb{E}_{x, z \sim p_{real}, p_z} \|E(G(O_i(z))) - O_i(z)\|_1. \quad (5)$$

**Theorem 2** (Inverse Distribution). *Suppose the cumulative distribution function of $O_i(z)$ is $q_i$. For any given positive real number $\epsilon$, there exist a $\delta > 0$ such that if $\mathcal{L}_{recon} + \mathcal{L}_{IV} \leq \delta$, then $\forall i, j \in [k]$, $\sup_r \|q_i(r) - q_j(r)\| \leq \epsilon$.*

The proof is in A.2.

**Adversarial loss**    The intervention loss and reconstruction loss can be added as regularization terms to the adversarial loss in many GAN models, e.g., the binary cross entropy loss in vanilla GAN and the least square loss in LSGAN. In the experiments, we use LSGAN (Mao et al., 2016) and DCGAN (Radford et al., 2015) as our base models, and name the resulting IVGAN models IVLSGAN and IVDCGAN respectively.

Now that we have introduced the essential components in the objective of IVGAN, we can write the loss function of the entire model:

$$\mathcal{L}_{model} = \mathcal{L}_{Adv} + \lambda\mathcal{L}_{recon} + \mu\mathcal{L}_{IV}, \quad (6)$$

where $\lambda$ and $\mu$ are the regularization coefficients for the reconstruction loss and the intervention loss respectively. We summarize the training procedure in Algorithm 1. A diagram of the full workflow of our framework can be found in Figire 3.

Figure 3: Full workflow of our approach.

## 4 EXPERIMENTS

In this section we conduct a series of experiments to study IVGAN from multiple aspects. First we evaluate IVGAN's performance on standard real-world datasets. Then we show IVGAN's ability to tackle the mode collapse problem on the stacked MNIST dataset. Finally, through an ablation study we investigate the performance of our proposed method under different settings of hyperparameters and demonstrate the effectiveness of the intervention loss.

We implement our models using PyTorch (Paszke et al., 2019) with the Adam optimizer (Kingma & Ba, 2015). Network architectures are fairly chosen among the baseline methods and IVGAN (see Table B.1 in the appendix for more details). The classifier we use to compute the intervention loss shares the parameters with the discriminator except for the output layer. All input images are resized to have $64 \times 64$ pixels. We use 100-dimensional standard Gaussian distribution as the prior $p_z$. We deploy the instance noise technique as in (Jenni & Favaro, 2019). One may check Appendix B.2 for detailed hyperparameter settings. All experiments are run on one single NVIDIA RTX 2080Ti GPU. Although IVGAN introduces extra computational complexities to the original framework of GANs, the training cost of IVGAN is within an acceptable range[1] due to the application of strategies like parameter sharing.

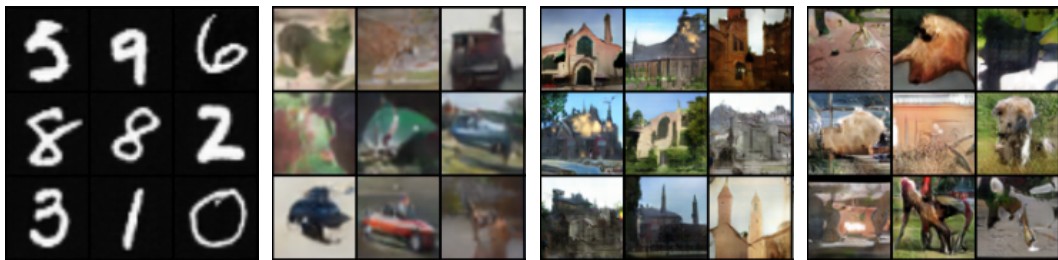

Figure 4: Random samples of generated images on MNIST, CIFAR-10, LSUN and STL-10.

**Real-world datasets experiments**   We first test IVGAN on four standard real-world datasets, including CIFAR-10 (Krizhevsky, 2009), MNIST (Lecun et al., 1998), STL-10 (Coates et al., 2011), and a subclass named "church_outdoor" of the LSUN dataset (Yu et al., 2015), to investigate its training stability and quality of the generated images. We use the *Frèchet Inception Distance* (FID) (Heusel et al., 2017) to measure the performance of all methods.

The FID results are listed in Table 1, and the training curves of the baseline methods and IVGAN on four different datasets are shown in Figure 5. We see that on each datasets, the IVGAN counterparts obtain better FID scores than their corresponding baselines. Moreover, the figure of training curves also suggests the learning processes of IVDCGAN and IVLSGAN are smoother and steadier compared to DCGAN, LSGAN or MRGAN (Che et al., 2016), and converge faster than WGAN-GP. Samples of generated images on all datasets are presented in Figure 4.

**Stacked MNIST experiments**   The original MNIST dataset contains 70K images of $28 \times 28$ handwritten digits. Following the same approaches in Metz et al. (2017); Srivastava et al. (2017); Lin et al. (2018), we increase the number of modes of the dataset from 10 to $1000 = 10 \times 10 \times 10$ by

---

[1]Empirically IVGANs are approximately 2 times slower than their corresponding baseline methods.

Table 1: Minimum of FIDs on different Datasets. The FID results are calculated every 10 epochs, and are averaged over five independent runs. Lower is better.

| Methods | MNIST | CIFAR10 | LSUN (Church_outdoor) | STL-10 |
|---|---|---|---|---|
| DCGAN | $9.87 \pm 1.18$ | $37.30 \pm 2.93$ | $23.18 \pm 0.98$ | $45.13 \pm 0.95$ |
| LSGAN | $12.06 \pm 0.88$ | $34.43 \pm 2.65$ | $29.53 \pm 1.10$ | $57.39 \pm 1.61$ |
| WGAN-GP | $10.83 \pm 0.66$ | $40.03 \pm 1.62$ | $26.08 \pm 0.60$ | $48.52 \pm 0.61$ |
| MRGAN | $8.52 \pm 1.19$ | $32.52 \pm 2.89$ | $21.03 \pm 0.90$ | $44.09 \pm 1.56$ |
| IVDCGAN | $\mathbf{8.27} \pm 1.25$ | $30.19 \pm 1.40$ | $20.30 \pm 0.80$ | $42.86 \pm 1.37$ |
| IVLSGAN | $8.33 \pm 0.25$ | $\mathbf{27.58} \pm 0.79$ | $\mathbf{18.99} \pm 0.88$ | $\mathbf{41.85} \pm 1.17$ |

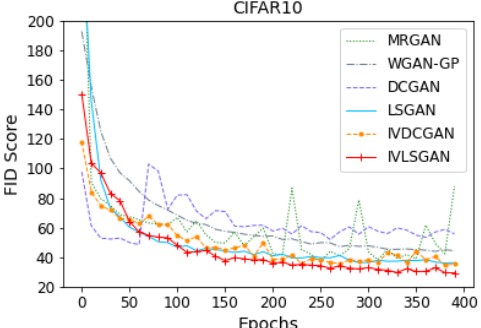 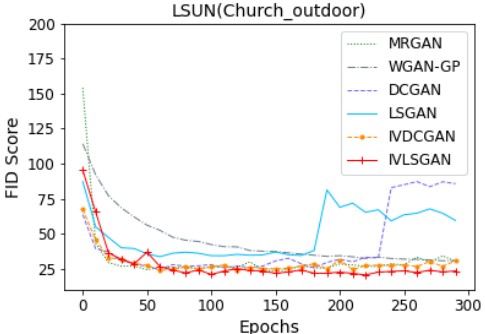

Figure 5: Training curves of different methods in terms of FID on different datasets, averaged over five runs. **Left**: CIFAR10. **Right**: Church Outdoors. Note that the raise of curves in the later stage may indicate mode collapse.

stacking three random MNIST images into a $28 \times 28 \times 3$ RGB image. The metric we use to evaluate a model's robustness to mode collapse problem is the number of modes captured by the model, as well as the KL divergence between the generated distribution over modes and the true uniform distribution. The mode of a generated imaged is found from a pre-trained MNIST digit classifier.

Our results are shown in Table 2. It can be seen that our model works very well to prevent the mode collapse problem. Both IVLSGAN and IVDCGAN are able to reach all 1,000 modes and greatly outperforms early approaches to mitigate mode collapse, such as VEEGAN (Srivastava et al., 2017), and Unrolled GAN (Metz et al., 2017). Moreover, the performance of our model is also comparable to method that is proposed more recently, such as the PacDCGAN (Lin et al., 2018). Figure 6 shows images generated randomly by our model as well as the baseline methods.

Table 2: Results of our stacked MNIST experiments. The first four rows are directly copied from (Lin et al., 2018) and (Srivastava et al., 2017). And the last three rows are obtained after training each model for 100K iterations, respectively.

| | Modes | KL Divergence |
|---|---|---|
| DCGAN | 78.9 | 4.50 |
| VEEGAN | 150.0 | 2.95 |
| Unrolled GAN | 48.7 | 4.32 |
| PacDCGAN | 1000 | 0.06 |
| LSGAN | 53 | 3.88 |
| IVLSGAN | 1000 | 0.07 |
| IVDCGAN | 1000 | 0.08 |

**Ablation study** Our ablation study is conducted on the CIFAR-10 dataset. First, we show the effectiveness of the intervention loss. We consider two cases, IVLSGAN without the intervention loss ($\mu = 0$), and standard IVLSGAN ($\mu = 0.5$). From Figure 7 we can find that the intervention loss makes the training process much smoother and leads to a lower FID score in the end.

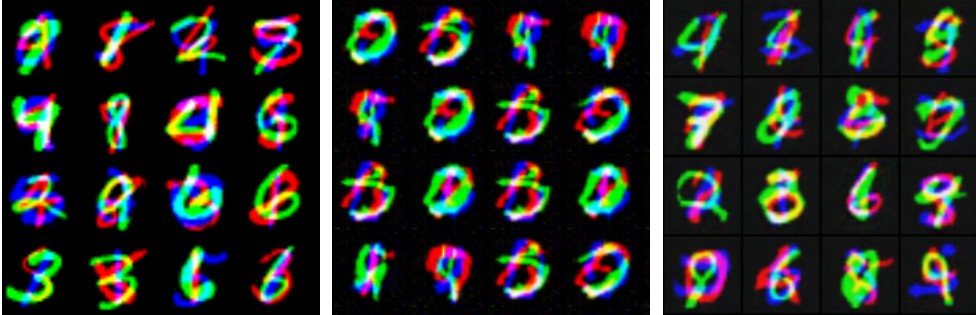

Figure 6: Sampled images on the stacked MNIST dataset. **Left**: Ground-truth. **Middle**: LSGAN. **Right**: IVLSGAN. Images generated by our method are more diverse.

We also investigate the performance of our model using different number of blocks for the block substitution interventions and different regularization coefficients for the intervention loss. The results are presented in Table 3. It can be noticed that to some extent our models' performance is not sensitive to the choice of hyperparameters and performs well under several different hyperparameter settings. However, when the number of blocks or the scale of IV loss becomes too large the performance of our model gets worse.

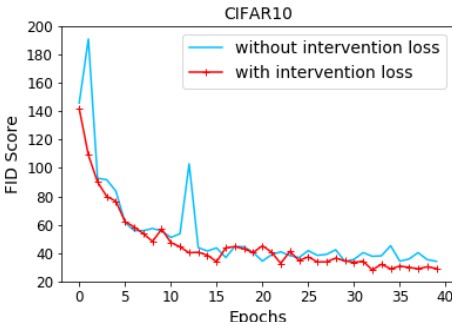

Figure 7: Training curve of IVLSGAN, with and without the intervention loss.

Table 3: Minimum FID scores of IVLSGAN under different hyperparameter settings on the CIFAR10 dataset, calculated every 10 epochs.

|  |  | FID score |
|---|---|---|
| $\mu = 0.5$ | $k = 2$ | 29.2 |
| $\mu = 0.5$ | $k = 4$ | 28.2 |
| $\mu = 0.5$ | $k = 10$ | 41.2 |
| $\mu = 0.5$ | $k = 20$ | 36.1 |
| $\mu = 0$ | $k = 4$ | 34.5 |
| $\mu = 0.25$ | $k = 4$ | 29.6 |
| $\mu = 0.5$ | $k = 4$ | 28.2 |
| $\mu = 1$ | $k = 4$ | 39.7 |

## 5 CONCLUSION

We have presented a novel model, intervention GAN (IVGAN), to stabilize the training process of GAN and alleviate the mode collapse problem. By introducing auxiliary Gaussian invariant interventions to the latent space of real images and feeding these perturbed latent representations into the generator, we have created intermediate distributions that interpolate between the generated distribution of GAN and the data distribution. The intervention loss based on these auxiliary intervened distributions, together with the reconstruction loss, are added as regularizers to the objective to provide more informative gradients for the generator, significantly improving GAN's training stability and alleviating the mode collapse problem as well.

We have conducted a detailed theoretical analysis of our proposed approach, and illustrated the advantage of the proposed intervention loss on a toy example. Experiments on both real-world datasets and the stacked MNIST dataset demonstrate that, compared to the baseline methods, IVGAN variants are stabler and smoother during training, and are able to generate images of higher quality (achieving state-of-the-art FID scores) and diversity. We believe that our proposed approach can also be applied to other generative models such as Adversarial Autoencoders (Makhzani et al., 2015), which we leave to future work.

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

# A PROOFS

## A.1 PROOF OF THEOREM 1

*Proof.* The conditional probability of $X'$ given label can be written as $\mathbb{P}(X'|e_i) = p_i(X')$, so further $\mathbb{P}(X', e_i) = \frac{1}{k} p_i$. And we denote the marginal distribution of $x$ as $p(x) = \frac{1}{k} \sum_{i=1}^{k} p_i(x)$. Cause the activation function at the output layer of the classifier is softmax, we can rewrite the loss function into a more explicit form:

$$V_{class}(f) = \mathbb{E}_{i \sim \mathcal{U}[k]} \mathbb{E}_{x' \sim p_i} - e_i^{\mathrm{T}} \log f(x') = \mathbb{E}_{i \sim \mathcal{U}[k]} \mathbb{E}_{x' \sim p_i} - \log f_i(x)$$

$$= \frac{1}{k} \int \sum_{i=1}^{k} -p_i(x) \log f_i(x) dx = \int p(x) \left\{ - \sum_{i=1}^{k} p(e_i|x) \log f_i(x) \right\} dx.$$

Let $g_i(x) = \frac{f_i(x)}{p(e_i|x)}$, then $\sum_{i=1}^{k} p(e_i|x)g_i(x) = 1$. And notice that $\sum_{i=1}^{k} p(e_i|x) = 1$. By Jensen's inequality, we have:

$$\sum_{i=1}^{k} -p(e_i|x) \log f_i(x) = \sum_{i=1}^{k} -p(e_i|x) \log[g_i(x)p(e_i|x)]$$

$$= \sum_{i=1}^{k} -p(e_i|x) \log g_i(x) + H(p(\cdot|x)) \geq \log \sum_{i=1}^{k} p(e_i|x)g_i(x) + H(p_i(\cdot|x))$$

$$= \log 1 + H(p(\cdot|x)) = H(p(\cdot|x)).$$

And $V_{class}(f^*) = \int p(x)H(p_i(\cdot|x))dx$ if and only if $g_i^*(x) = g_j^*(x)$ for any $i \neq j$, which means that $\frac{f_i^*(x)}{p(e_i|x)} = r \quad \forall i \in [k]$, where $r \in \mathbb{R}$. Notice that $\sum_{i=1}^{k} f_i^*(x) = 1$, it is not difficult to get that $f_i^*(x) = p(e_i|x)$. The loss function becomes

$$\frac{1}{k} \int \sum_{i=1}^{k} -p_i(x) \log p(e_i|x)dx = -H(x) + \sum_{i=1}^{k} \frac{1}{k}H(p_i) + \log k \tag{7}$$

$$= -JS(p_1, p_2, ..., p_k) + \log k$$

$\square$

### A.2 PROOF OF THEOREM 2

*Proof.* According to Theorem 1, for a given real number $\epsilon_1$, we can find another $\delta_1$, when intervention loss is less than $\delta_1$, the distance between $p_i$ and $p_j$ under the measurement of JS-divergence is less than $\epsilon_1$. And because JS-divergence and Total Variance distance (TV) are equivalent in the sense of convergence. So we can bound the TV-distance between $p_i$ and $p_j$ by their JS-divergence. Which means that $\int |p_i - p_j|dx \leq \epsilon_0$ when the intervention loss is less than $\epsilon_1$ (we can according to the $\epsilon_0$ to finding the appropriate $\epsilon_1$). Using this conclusion we can deduce $|P(E(G(O_i(z))) \leq r) - P(E(G(O_j(z))) \leq r)| \leq \epsilon_0$, where $r$ is an arbitrary vector in $\mathbb{R}^d$. Further, we have:

$$|P(O_i(z) \leq r) - P(O_j(z) \leq r)| \leq |P(O_i(z) \leq r; \|O_i(z) - E(G(O_i(z)))\| > \delta)|$$
$$+ |P(O_j(z) \leq r; \|O_j(z) - E(G(O_j(z)))\| > \delta)| + |P(O_i(z) \leq r; \|O_i(z) - E(G(O_i(z)))\| \leq \delta)$$
$$- P(O_j(z) \leq r; \|O_j(z) - E(G(O_j(z)))\| \leq \delta)| \tag{8}$$

We control the three terms on the right side of the inequality sign respectively.

$$P(O_i(z) \leq r; \|O_i(z) - E(G(O_i(z)))\| > \delta)$$
$$\leq P(\|O_i(z) - E(G(O_i(z)))\| > \delta) \leq \frac{\mathbb{E}\|O_i(z) - E(G(O_i(z)))\|}{\delta} \tag{9}$$

And the last term can be bounded by the reconstruction loss. The same trick can be used on $P(O_j(z) \leq r; \|O_j(z) - E(G(O_j(z)))\| > \delta)$. Moreover, we have

$$P(E(G(O_i(z))) \leq r - \delta) - P(\|O_i(z) - E(G(O_i(z)))\| > \delta)$$
$$\leq P(O_i(z) \leq r; \|O_i(z) - E(G(O_i(z)))\| \leq \delta) \leq P(E(G(O_i(z))) \leq r + \delta) \tag{10}$$

Notice that $\lim_{\delta \to 0} P(E(G(O_i(z))) \leq r \pm \delta) = P(E(G(O_i(z))) \leq r)$. Let $s_i(r, \delta) = |P(E(G(O_i(z))) \leq r \pm \delta)) - P(E(G(O_i(z))) \leq r)|$ then the last term of inequalityA.2 can be bounded as:

$$|P(O_i(z) \leq r; \|O_i(z) - E(G(O_i(z)))\| \leq \delta) - P(O_j(z) \leq r; \|O_j(z) - E(G(O_j(z)))\| \leq \delta)|$$
$$\leq |P(E(G(O_i(z))) \leq r) - P(E(G(O_j(z))) \leq r)| + P(\|O_i(z) - E(G(O_i(z)))\| > \delta)$$
$$+ s_i(r, \delta) + s_j(r, \delta) \tag{11}$$

Every term on the right hand of the inequality can be controlled close to 0 by the inequalities mentioned above. $\square$

## B  EXPERIMENTAL DETAILS

### B.1  NETWORK ARCHITECTURES

Table 4: The NN architecture used by us, where CONV denotes the convolutional la yer, TCONV denotes the transposed convolutional layer, FC denotes the fully-connected layer, BN denotes the batch normalization layer, and (K4, S1, O512) denotes a layer with kernel of size 4, stride 1, and 512 output channels.

| D | G | E |
|---|---|---|
| INPUT 64×64×3 | INPUT z | INPUT 64×64×3 |
| CONV(K4, S2, O64) | TCONV(K4, S1, O512) | CONV(K4, S2, O64) |
| BN, LeakyReLU | BN, ReLU | LeakyReLU |
| CONV(K4, S2, O128) | TCONV(K4, S2, O256) | CONV(K4, S2, O128) |
| BN, LeakyReLU | BN, ReLU | BN, LeakyReLU |
| CONV(K4, S2, O256) | TCONV(K4, S2, O128) | CONV(K4, S2, O256) |
| BN, LeakyReLU | BN, ReLU | BN, LeakyReLU |
| CONV(K4, S2, O512) | TCONV(K4, S2, O64) | CONV(K4, S2, O512) |
| BN, LeakyReLU | BN, ReLU | BN, LeakyReLU |
| FC(O1) | TCONV(K4, S2, O3) | CONV(K4, S2, O100) |
| LOSS | Tanh | BN |

### B.2  HYPERPARAMETER SETTINGS

Table 5: Detailed hyperparameter settings in our experiments.

| | Settings of the Adam optimizer | | | | | | Other Hyperparameters | | |
|---|---|---|---|---|---|---|---|---|---|
| | $lr_D$ | $lr_G$ | $lr_E$ | BatchSize | $\beta_1$ | $\beta_2$ | | | |
| DCGAN | 0.0001 | 0.0001 | | 64 | 0.5 | 0.999 | | | |
| LSGAN | 0.0002 | 0.0002 | | 64 | 0.5 | 0.999 | | | |
| WGANGP | 0.0001 | 0.0001 | | 64 | 0.5 | 0.9 | $n_{dis} = 5$ | $\lambda_{GP} = 10$ | |
| MRGAN | 0.0002 | 0.0002 | 0.01 | 64 | 0.5 | 0.999 | $\lambda_1 = 0.25$ | $\lambda_2 = 0.25$ | |
| IVDCGAN | 0.0002 | 0.0002 | 0.005 | 64 | 0.5 | 0.999 | $\mu = 0.5$ | $\lambda = 0.25$ | $k = 4$ |
| IVLSGAN | 0.0002 | 0.0002 | 0.005 | 64 | 0.5 | 0.999 | $\mu = 0.5$ | $\lambda = 0.25$ | $k = 4$ |

## C  RELATED WORK

In order to address GAN's unstable training and mode missing problems, many researchers have turned their attention to the latent representations of samples. Makhzani et al. (2015) proposed the *Adversarial Autoencoder* (AAE). As its name suggests, AAE is essentially a probabilistic autoencoder based on the framework of GANs. Unlike classical GAN models, in the setting of AAE the discriminator's task is to distinguish the latent representations of real images that are generated by an encoder network from Gaussian noise. And the generator and the encoder are trained to fool the discriminator as well as reconstruct the input image from the encoded representations. However, the generator can only be trained by fitting the reverse of the encoder and cannot get any information from the latent representation.

The VAE-GAN (Larsen et al., 2015) combines the objective function from a VAE model with a GAN and utilizes the learned features in the discriminator for better image similarity metrics, which is of great help for the sample visual fidelity. Considering the opposite perspective, Che et al. (2016) claim that the whole learning process of a generative model can be divided into the manifold learning phase and the diffusion learning phase. And the former one is considered to be the source of the mode missing problem. (Che et al., 2016) then proposed *Mode Regularized Generative Adversarial*

*Nets* which introduce a reconstruction loss term to the training target of GAN to penalize the missing modes. It is shown that it actually ameliorates GAN's 'mode missing'-prone weakness to some extent. However, both of them fail to fully excavate the impact of the interaction between VAEs and GANs.

Kim & Mnih (2018) proposed Factor VAE where a regularization term called total correlation penalty is added to the traditional VAE loss. The total correlation is essentially the Kullback-Leibler divergence between the joint distribution $p(z_1, z_2, \ldots, z_d)$ and the product of marginal distribution $p(z_i)$. Because the closed forms of these two distribution are unavailable, Factor VAE uses adversarial training to approximate the likelihood ratio.

