# OpenReview forum: "Intervention Generative Adversarial Nets"
_ICLR.cc/2021/Conference — Reject_

### Official Review · AnonReviewer4 · 2020-10-22
**Clear Reject**

**Rating:** 3
**Confidence:** 5

**Review:**

Intervention Generative Adversarial Nets

Summary

This paper proposes a GAN training procedure where images are encoded into latents, which are perturbed and passed through a generator/decoder, and the discriminator objective is augmented to encourage it to classify the perturbation. Results are presented for generative image modeling on several datasets and compared against DCGAN, LSGAN, and WGAN-GP.

My take

This paper is another one of a common variant of GAN papers which makes small changes to the objective and training setup and shows small gains relative to old baselines on small-scale tasks. This paper’s empirical results are unfortunately lacking and there are no substantial or new theoretical insights, making this paper a clear reject.

Detailed review

Significance

This paper proposes yet another GAN training procedure, based on an encoder-decoder-discriminator setup (as has been done many times before, especially in VAE-GAN setups). The change to the vanilla training setup is effectively quite small, but introduces a 100% compute overhead relative to its baselines. My primary complaint with this paper is that for such a small change to be of interest to the community (especially given its tremendous compute cost and any added implementation complexity) there must be substantial evidence that doing so is beneficial, and the empirical results in this paper are quite weak.

This paper compares against old baselines and obtains numbers which do not justify the computational overhead even relative to these old baselines. This paper should include comparison against more recent work if it wishes to prove its relevance--a quick look at an online leaderboard shows CIFAR-10 (https://paperswithcode.com/sota/image-generation-on-cifar-10) has tons of recent work which the authors could compare to or build off of. WGAN-GP is far from state-of-the-art on any of the tested benchmarks, and that the authors have made any sort of SOTA claims is misleading, and not acceptable.

Clarity

The paper is reasonably well written and easy to follow.

Methodological soundness

The paper’s methodology is reasonably sound, and not explicitly cause for concern.

Originality

This paper is not especially original. There are dozens, if not hundreds, of GAN training papers which introduce small changes to the (now 6 year old) vanilla GAN or VAE-GAN objective, and there are no ideas in this paper which I would consider espeically new or novel.

Misc

-The authors seem to conflate mode collapse (when large regions of Z map to small regions of G(Z)) with mode dropping (when modes in X are not represented in G(Z)). These are two separate phenomena and while they co-occur, they should not be considered the same.

-There is no discussion of previous work on “intervention,” or any of the related concepts. Being a critical element of the proposed method, this should be discussed in more detail (rather than just focusing on related GAN work).

-There is an obvious connection to InfoGAN, which should be cited.

---

### Official Review · AnonReviewer1 · 2020-10-26
**Review of Intervention Generative Adversarial Nets**

**Rating:** 6
**Confidence:** 3

**Review:**

The paper proposes a method for stabilizing the training of GAN as well as overcoming the problem of mode collapse by optimizing several auxiliary models. The first step is to learn a latent space using an autoencoder. Then, this latent space is "intervened" by a predefined set of $K$ transformations to generate a set of distributions $p_k$. A classifier is then taught to distinguish between $p_k$. Eventually, the weights of the classifier are shared with those of the discriminator network to produce the desired stabilization/diversification effect. In other words, the authors propose to stabilize GANs by intervening with the discriminator. This is done by sharing its weights with a classifier that trains on a perturbed latent distribution that is somehow related to the original problem via the prior assumption imposed.

Throughout the paper, the authors use a \textbf{block substitution} transformation as the intervening transformation. In this case, the kth transformation replaces the kth block in the embedding $e(X)$ of a sample $X$ with Gaussian noise. The authors also present a theorem relating the value of the classification loss function at the optimal solution to the Jensen Shannon divergence between the intervened distributions.
Finally, the authors apply their approach to generate images using several datasets (MNIST-10, CIFAR, LSUN, STL-10). Figure 5 demonstrates that the approach outperforms the previous methods.

Issues:
A. The use of block transformations is not well motivated. Since it seems like an arbitrary design choice, I would appreciate it if the authors could justify its use.
B. The approach seems quite cumbersome and adds non-negligible overhead compared to the baseline. Furthermore, looking at Figure 7, the gain appears marginal.
C. The authors introduce a high order extension of JS divergence, denoted Multi-distribution JS divergence, and show that standard JS divergence is a special case for $k=2$ and $\pi_1=\pi_2=0.5$. The authors then use this quantity in Theorem 1 to relate the intervention loss to JS divergence. However, it is hard for me to see this logical move as valid proof of anything. Although the high-order JS divergence is intuitive, it is still a subjective quantity that the authors define and then use to make a formal statement. I think it's acceptable to use the high-order JS divergence as intuition about the intervention loss, but problematic to use it as proof of anything.
D. It is difficult for me to evaluate "diversity" based on a few samples, particularly in the "stacked" MNIST images my eyes cannot read.

Personal note:
The reviewer admits his unfamiliarity with recent research in this field. Therefore, he cannot assess the novel nature of the method nor its significance. I see this as borderline work. However, with no such score available, I gave it a slightly positive rating.

---

### Official Review · AnonReviewer2 · 2020-10-29
**Review for "Intervention Generative Adversarial Nets"**

**Rating:** 2
**Confidence:** 5

**Review:**

This paper introduces "Intervention GAN (IVGAN)", an algorithm proposed to stabilize the training process os GANs as well as alleviating the mode collapse problem. In IVGAN the authors feed O(x) to the generator with O different possible transformations and train the discriminator to also differentiate which transformation originated the sample.

The approach is extremely poorly motivated. Why transformations like this have anything to do with mode colapse or training stability is not at all at the level of this conference. This paper is literally written as "We came up with this method and we tried out whether it works or not, the conclusion is it doesn't really hurt in our limited experiments, sometimes it helps by an epsilon". This is really not up to the standards of ICLR, I would expect at least a tiny step towards solving a fundamental problem with a clear motivation and research path, or at the very least better experimental results.

There is no quantification of training stability, and the only qualitative experiment in that regard is figure 5, which doesn't really provide a strong argument for the model in my opinion. If they showed they could train faster difficult models (such as bigger ones like SGAN) or they could train models for new tasks where previous methods fail that would be better, but this is really quite underwhelming.

The only quantification for mode dropping is on mnist, a dataset that's widely accepted as useless in the current state of generative modelling. Furthermore, important baselines like wgangp are missing.

Furthermore, the language and the overclaiming of the paper is very strong: "our approach is able to stabilize GAN's training and improve the quality and diversity of generated samples as well". Do the authors really think that their approach was finaly able to stabilize GAN training? The way these sentences are written make it seem like IVGAN is as easy to train in imagenet as training a resnet with cross-entropy for supervised learning. It is simply unacceptable given the limited experiments (tiny datasets, small improvements, far from the current state of generative modelling).

Honestly, I feel very dissapointed about how this paper presents itself. Poorly motivated ideas with an epsilon improvement are not meant for top tier conferences, and the degree to what this paper overclaims is not acceptable either.

---

### Official Review · AnonReviewer3 · 2020-10-31

**Rating:** 7
**Confidence:** 4

**Review:**

This work introduces a regularization term in GANs to both improve training stability and remedy mode collapse.  The approach, as I understand it, is the follows:
* Real images are encoded into a latent representation (encoder learned jointly with separate loss)
* From an available set of perturbations or interventions, this latent representation is transformed. This work does a block substitution of a particular slice of the latent representation vector.
* The intervened latent representation is given to the generator which produces an intervened sample.
* A classifier determines which intervention the intervened sample belongs to.

The authors describe how this intervention loss is equivalent to a multi-JSD. This was a rather intricate algorithm and I think it’s generally well-explained. This paper demonstrates clear empirical improvements of the method on many benchmarks of interest. I therefore recommend acceptance.

Questions to authors:
* This work focuses on the block-intervention, however, does the nature of the intervention matter?
* Does this intervention loss have applicability outside of GANs?
* Are the p_real latent distributions approximately Gaussian? Does a mismatch with the latent distribution used to produce fake images impact performance?

Notes:
* Regarding the comment “Training GAN has been challenging, especially when the generated distribution and the data distribution are far away from each other. In such cases, the discriminator often struggles to provide useful information for the generator, leading to instability and mode collapse problems”, Fedus et al. (2017) showed non-overlapping data distributions are learnable by a non-saturating GAN. You may want to qualify this statement. In practice, we never use a *perfect* discriminator.
* Example 1 gave nice intuition on this complicated method.
* Nit: Remove subjective claims in your writing -- this will strengthen the presentation, e.g. “...via solid theoretical analysis...”.
* Nit: Using the word “novel” is redundant since you are proposing it, e.g. “In this paper we propose a novel technique for GANs”
* Nit: Recommend converting your Figures from PNG to PDF to improve the image quality, especially for online viewers.

---

### Decision · Program_Chairs · 2021-01-07
**Final Decision**

**Decision:**

Reject

**Comment:**

Nominally, the scores on this paper were pretty split.
In reality, I concur with the 2 and the 3.
The 6 acknowledges being unfamiliar w/ the GAN literature, and I think the 7 is being too permissive about the baselines.

The empirical evaluation here is simply not up to par for a major machine learning conference.
As reviewers have mentioned, the baselines are out of date, and even then the improvements are marginal.
It's totally fine to have a marginal improvement if the proposed technique is very new and interesting and the baselines
are taken seriously, but unfortunately I don't believe that's the case here.
Thus, I recommend rejection.